# Prevalence of SARS-COV-2 positivity in 516 German intensive care and emergency physicians studied by seroprevalence of antibodies National Covid Survey Germany (NAT-COV-SURV)

**Detlef Kindgen-Milles**[1]*, **Timo Brandenburger**[1], **Julian F. W. Braun**[2], **Corvin Cleff**[3], **Kian Moussazadeh**[4], **Ingo Mrosewski**[5], **Jörg Timm**[6], **Dietmar Wetzchewald**[2]

**1** Dept. of Anesthesiology, University Hospital Duesseldorf, Heinrich-Heine University Duesseldorf, Duesseldorf, Germany, **2** Dept. of Emergency Medicine, University Witten/Herdecke, Arnsberg, Germany, **3** Department of Anaesthesiology and Intensive Care Medicine, Faculty of Medicine and University Hospital Cologne, University of Cologne, Cologne, German, **4** Dept. of Anesthesiology & Intensive Care Sana Hospital Duisburg, Duisburg, Germany, **5** Department of Laboratory Medicine, MDI Limbach Berlin GmbH, Berlin, Germany, **6** Institute of Virology, University Hospital Duesseldorf, Heinrich-Heine University Duesseldorf, Duesseldorf, Germany

* kindgen-milles@med.uni-duesseldorf.de

**Data Availability Statement:** According to german federal and state jurisdiction (DGSVO,

## Abstract

Healthcare personnel are at risk to aquire the corona virus disease 2019 (COVID-19) caused by the severe acute respiratory syndrome coronavirus 2 (SARS-CoV-2). We evaluated the prevalence of SARS-CoV-2 antibodies and positive nasopharyngeal reverse transcriptase polymerase-chain reaction (RT-PCR) tests in German intensive care and emergency physicians. Physicians attending intensive care and emergency medicine training courses between June 16th and July 2nd 2020 answered a questionnaire and were screened for SARS-CoV-2 antibodies via automated electrochemiluminiscence immunoassay. We recruited 516 physicians from all parts of Germany, 445/516 (86%) worked in high risk areas, and 379/516 (73%) had treated patients with COVID-19. The overall positive rate was 18/516 (3.5%), 16/18 (89%) had antibodies against SARS-COV-2, another 2 reported previous positive RT-PCR results although antibody testing was negative. Of those positive, 7/18 (39%) were unaware of their infection. A stay abroad was stated by 173/498 (35%), mostly in Europe. 87/516 (17%) reported a febrile respiratory infection after January 1st 2020 which was related to SARS-CoV-2 in 4/87 (4.6%). Contact to COVID-19 positive relatives at home was stated by 22/502 (4.4%). This was the only significant risk factor for Covid-19 infection (Fisher´s exact test, $p = 0.0005$). N95 masks and eye protection devices were available for 87% and 73%, respectively. A total of 254/502 (51%) had been vaccinated against seasonal influenza. The overall SARS-CoV-2 infection rate of german physicians from intensive care and emergency medicine was low compared to reports from other countries and settings. This finding may be explained by the fact that the German health care system was not overwhelmed by the first wave of the SARS-CoV-2 pandemic.

Datenschutzgrundverordnung) data are protected and cannot be shared publicly. The approval by the committee of medical ethics at Heinrich-Heine University Duesseldorf (*2020-1034) did not allow transfer of data outside of the university. However, data can be made available upon request. For any data request researchers may contact: (1) Committee of Medical Ethics of Heinrich-Heine University, Medical Faculty Heinrich-Heine University Duesseldorf, Moorenstr. 5, D-40225 Duesseldorf; ethikkommission@med.uni-duesseldorf.de, or (2) the data protection commissioner of the university hospital: Datenschutzbeauftragter, University Hospital Duesseldorf, Moorenstr. 5, D-40225 Duesseldorf; datenschutz@med.uni-duesseldorf.de.

**Funding:** The authors received no specific funding for this work.

**Competing interests:** Dr. Ingo Mrosewski is employed by the MDI Limbach Berlin GmbH, a commercial clinical laboratory. The MDI Limbach GmbH did not play a role in the study design, data collection and analysis, decision to publish, or preparation of the manuscript. It did not provide specific financial support but all costs for laboratory analysis were paid for by the Dept. of Emergency Medicine at University of Herdecke. This affiliation does not alter our adherence to PLOS ONE policies on sharing of data and materials. There are no patents, products in development or marketed products associated with this research to declare. All other authors have nothing to declare.

## Introduction

The first cases of severe acute respiratory coronavirus 2 (SARS-CoV-2) infection were reported in December 2019. Meanwhile, corona virus disease 2019 (COVID-19) has become a pandemic [1]. Medical personnel was exposed to the risk of virus transmission and subsequent severe illness. However, many infected persons may present no or only mild symptoms so the true spread of the disease is unknown. This applies also to healthcare workers (HCW) in whom prevalence studies are not performed routinely. Medical staff at high risk are physicians working in intensive care units and emergency rooms, in the operating theatre, or as emergency physicians in prehospital settings. In these groups, virus transmission and a large number of COVID-19 associated physican deaths have been reported [2].

In a symptomatic disease course, SARS-CoV-2 can be detected by reverse transcriptase polymerase-chain reaction (RT-PCR). In healthy patients with unknown COVID-19 exposure or in reconvalescents, the detection of antibodies against viral targets can be used [3]. Thus, seroprevalence studies are important for determing the extent of an outbreak, either in a defined area or in a defined at risk group.

At present, most prevalence studies were performed after outbreaks in COVID-19 hotspot areas or in single hospital settings. As of yet, no widespread information on the rate of SARS-CoV-2 infection in German healthcare workers, and especially in physicians working in high-risk areas, is available. Therefore, we performed this modified point prevalence study to investigate the prevalence of SARS-CoV-2 in intensive care and emergency physicians from all over Germany.

## Materials and methods

### Study design and participants

The study was approved by the committee of medical ethics at Heinrich-Heine University Duesseldorf (*2020–1034) and registered at clinicaltrials (NCT04459312). Written informed consent was obtained from all participants.

From June 16th to July 2nd 2020 we recruited physicians from all over Germany who attended certified registered training courses for intensive care or emergency medicine in the city of Arnsberg in Northrhine-Westfalia. These courses are organized by the "Working Group Intensive Care Medicine" (Arbeitsgemeinschaft Intensivmedizin e.V.), take place 11–12 times every year since 1991, and are attended by approx. 2800 physicians per year. In 2020, the spring courses were postponed due to the corona pandemic and took place in June, so that a high number of physicians was recruitable in a short period of time.

All participants answered a clinical questionnaire and from all participants 8 ml blood were drawn for SARS-CoV-2 antibody detection. All questionnaires and blood samples were given matching numbers and blinded. In some participants RT-PCR testing had been performed independent from this study. Participants were asked whether such tests had been performed and to provide the results of this tests, i.e. positivity or a negative result.

### Sample collection, transportation and storage

Blood samples were collected in serum monovettes (SARSTEDT AG & Co. KG, Nümbrecht, Germany) via direct venipuncture. After complete coagulation, the samples were centrifuged at 2000 g at room temperature for 10 minutes. Supernatants were transferred to secondary polypropylene tubes (SARSTEDT AG & Co. KG, Nümbrecht, Germany) and stored at 4–8˚C. After serological testing, the aliquots were frozen and stored at -20˚C.

### SARS-CoV-2 antibody detection

Serological testing was performed within 3–7 days after sample collection. Antibodies directed against SARS-CoV-2 were detected by automated, CE certified electrochemiluminiscence immunoassay (ECLIA) (Reference number 09203079190, Roche Diagnostics GmbH, Mannheim, Germany). The Elecsys anti-SARS-CoV-2 assay is a combined qualitative test for immunoglobulin (Ig) G, M and A antibodies against the nucleocapsid (N) protein of SARS-CoV-2. A cutoff-index $\geq 1.0$ denoted positive samples (for test details see: Roche Diagnostics GmbH. https://primeservices.roche.com/eLD_SF/de/de/Documents/GetDocument?documentId=cccdd6a2-de95-ea11-fc90-005056a71a5d).

Positive samples were further evaluated via semiquantitative, automated anti-SARS-CoV-2 IgG and IgA enzyme-linked immunosorbent assay (ELISA) (Reference number EI 2606–9601 G and EI 2606–9601 A, EUROIMMUN AG, Lübeck, Germany). The ELISA plates have a capacity of 96 wells and are coated with recombinant expressed spike 1 (S1) domain glycoprotein of SARS-CoV-2. The overall diagnostic specificity for the anti-SARS-CoV-2 IgG ELISA is given as 99.3% and for the anti-SARS-CoV-2 IgA ELISA as 88.2–92.4%. The data sheets report cross-reactivity with SARS-CoV-1, but not with MERS-CoV, HCoV-229E, HCoV-NL63, HCoVHKU-1 or HCoVOC43 virus.

For semiquantitative results, a ratio of the optical density (OD) of each sample to the reading of the calibrator, included in the kit, was automatically calculated according to the formula: OD ratio = OD of serum sample/OD of calibrator. A ratio $< 0.8$ denoted negative, $0.8 - < 1.1$ borderline and $\geq 1.1$ positive results.

### Statistical analysis

This is an observational study designed to describe the prevalence of SARS-COV2 infection in German physicians. We tested for group differences using the two-sided Fisher´s exact test. A threshold of $p < 0.05$ was set for statistical significance.

## Results

### Population

From June 16$^{th}$ to July 2$^{nd}$ 2020 539 physicians attended the courses of whom 516 agreed to participate in this study. Most participants, i.e. 261/516, came from Northrhine-Westfalia which is the largest federal state in Germany with approx. 18 million inhabitants followed by Hessen (51; population 6 million), Bavaria (42; population 13 million) Baden-Wuerttemberg (34; population 11 million), Lower-Saxony (36; population 8 million), and the Berlin area (20, population 3.7 million).

Demographic data and physician´s affiliations are shown in Table 1. The gender distribution was equal, the mean age was 33.1 ± 6.5 years, and 56% of physicians were 30–39 years old. The mean work experience as physician was 5.3 ± 5.9 years, 85% of participants were specialist registrars and 10% were senior physicians or head of departments. The predominant specialist area was internal medicine (52%), followed by anaesthesiology (28%), surgery (13%), neurology (5%), and other faculties (2%).

Table 2 shows hospital and deployment data of all participants. Physicians came from hospitals of different sizes between less than 250 and more than 1000 beds, 20% of the participants worked in hospitals with more than 1000 beds thus representing tertiary centers and/or university hospitals. Most participants worked in different hospital settings but 454/516 (88%) worked in high risk areas including intensive care units, operating theatres, emergency

**Table 1. Study participants: Demographic data.**

| Characteristics (n = 516) | No. (%) |
|---|---|
| *Gender* | |
| Female | 258 (50%) |
| Male | 258 (50%) |
| *Age (years)* | |
| Mean ± SD | 31.1±6.5 |
| < 30 | 165 (32%) |
| 30–39 | 287 (56%) |
| 40–49 | 49 (9%) |
| 50–59 | 11 (2%) |
| ≥ 60 | 4 (1%) |
| *Position* | |
| Specialist registrar | 437 (84%) |
| Senior physician | 45 (9%) |
| Head of Dept. | 4 (1%) |
| Other | 30 (6%) |
| *Faculty* | |
| Surgery | 66 (13%) |
| Internal Medicine | 268 (52%) |
| Neurology | 23 (5%) |
| Anesthesiology | 145 (28%) |
| Other | 14 (2%) |
| *Years of clinical work* | |
| *0–5* | 365 (70%) |
| *6–9* | 76 (15%) |
| *≥ 10* | 75 (15%) |

departments and out-of-hospital physician-based emergency services, 379/516 (73%) had treated patients with proven SARS-CoV-2 infection.

## Seroprevalence of antibodies against SARS-CoV-2 and proven infection

We found antibodies against the N protein of SARS-CoV-2 in 16/516 (3.1%) participants. SARS-CoV-2 tests via nasopharyngeal swab and RT-PCR and/or antibody detection independent from this study, i.e. at a time prior to participation in this study, had been performed in 222/516 (43%). RT-PCR had confirmed SARS-CoV-2 infection in 2 participants (0.4%) who were negative in the Elecsys anti-SARS-CoV-2 assay. Thus, the prevalence of SARS-CoV-2 infection in our study population was 18/516 (3.5%). Of those with positive antibodies or a positive RT-PCR 11/18 (61%) were aware of their infection–mostly due to previous RT-PCR testing unrelated to this study—and 7/18 (39%) were not aware.

After positive Elecsys anti-SARS-CoV-2 assay, using the EUROIMMUN ELISA we detected in 13/16 (81%) IgA and in 14/16 (88%) IgG antibodies against the SARS-CoV-2 S1 protein.

## Travel history

498/516 participants answered the questions about travel history. Of those, 173/498 (35%) had stayed abroad after January 1st 2020 for a mean of 14 ± 24 days (Table 3). Physicians had visited 57 different countries. Most of the journeys took place in Europe (76%), followed by Asia (10%), Africa (5.9%), North America (3.6%), and Australasia (3.1%). We found no statistically

**Table 2. Hospital and deployment statistics.**

| Characteristics (n = 516) | No. (%) |
|---|---|
| *Treating patients with COVID-19* | |
| Yes | 379 (73%) |
| No | 118 (23%) |
| Unknown | 19 (4%) |
| *Hospital size (beds)* | |
| <250 | 92 (18%) |
| 250–500 | 163 (32%) |
| 501–750 | 102 (20%) |
| *751–1000* | 38 (7%) |
| ≥1001 | 103 (20%) |
| Not specified | 18 (3%) |
| *Deployment** | |
| Emergency room | 162 (31%) |
| Intensive care unit | 313 (61%) |
| General ward | 212 (41%) |
| Operating room | 129 (25%) |
| Emergency medical service | 44 (9%) |
| Other | 33 (6%) |
| *Available protective equipment** | |
| FFP2 mask | 363 (70%) |
| FFP3 mask | 176 (34%) |
| Protective goggles | 350 (68%) |
| Face shield | 210 (41%) |

significant correlation between stays abroad and risk of SARS-CoV-2 infection in this study (Fisher´s exact test; $p$ = 0.193).

## Febrile respiratory infection and home contacts to COVID-19 patients

87/499 (17%) reported a febrile respiratory tract infection after January 1st 2020. Among these, only 4/87 (4.6%) were seropositive for SARS-CoV-2. Among the 16 participants who were SARS-CoV2 positive, only 2/18 (11%) reported a previous febrile respiratory infection after January 1st 2020. We found no statistically significant correlation between a clinically apparent febrile respiratory infection and SAR-CoV-2 positivity (Fisher´s exact test; $p$ = 0.534) (Table 3).

Contact to patients with COVID-19 at home was reported by 22/502 (4.4%) physicians. Of those positive for SARS-CoV-2, 5/18 (28%) had contact to SARS-CoV-2 positive family

**Table 3. Risk factors for SARS-CoV-2 positivity.**

| | SARS-CoV-2 positive | SARS-CoV-2 negative | p |
|---|---|---|---|
| All | 18 (3.5%) | 498 (96.5%) | |
| Contact at home | 5 (27.7%) | 17 (3.4) | 0.0005 |
| Stay abroad | 8 (44.4%) | 318 (63.9%) | 0.193 |
| Feverish resp. Infection | 4 (22.2%) | 83 (17.3%) | 0.585 |
| Masks available | 16 (88.9%) | 437 (87.8%) | 0.884 |
| Face shields available | 15 (83.3%) | 432 (83.7%) | 0.965 |

members while 13/18 (72%) had not. Of those negative for SARS-CoV-2, 17/480 (3.5%) had contact to family members with COVID-19 and 463/480 (96%) had not. We found a statistically significant correlation between contact to SARS-CoV-2 positive family members and SARS-CoV-2 infection in our participants (Fisher´s exact test; $p = 0.0005$).

### Personal protective equipment

N95 masks (in Germany defined as FFP2 or FFP3, the latter having a higher filtering capacity) were available to most participants, i.e. FFP2 masks to 363/516 (70%) and FFP3 to 176/516 (34%). In some hospitals both types of masks were used. However, for 69/516 (13%) participants neither FFP2 nor FFP3 masks were available at some point in time. No difference was detected between physicians with SARS-CoV-2 infection, i.e. 2/18 (11%), and those without SARS-SoV-2 infection, i.e. 67/498 (13%). We found no statistically significant correlation between the availability of FFP2/FFP3 masks and SARS-CoV-2 infection (Fisher´s exact test; $p = 0.99$) (Table 3).

Protective goggles were available to 350/516 (68%) and face shields to 210/516 (41%). In some hospitals both devices were used. To 93/516 (18%) none of these devices was available at some point in time. No difference was detected for physicians with SARS-CoV-2 infection, i.e. 3/18 (17%) and those without SARS-CoV-2 infection, i.e. 90/498 (18%). We found no statistically significant correlation between the availability of eye protective devices and SARS-CoV-2 infection (Fisher´s exact test; $p = 0.99$).

### Vaccination against seasonal influenza

We asked participants whether they had been vaccinated against seasonal influenza in the 2019/2020 season. This question was answered by 500 participants and 254/502 (51%) had been vaccinated. We analyzed whether SARS-CoV-2 positivity was correlated to vaccination against seasonal influenza: 8/254 (3.1%) of those vaccinated against seasonal influenza and 8/246 (3.3%) of those not vaccinated were positive for SARS-CoV-2. The difference was not statistically significant.

### Discussion and conclusions

We found an overall positivity rate for either antibodies against SARS-CoV-2 or a positive RT-PCR test in 3.5.% of German physicians working in high-risk areas, namely intensive care units, emergency departments, operating theaters and out-of hospital physican-based emergency services.

The positive rate for SARS-CoV-2 infections (3.5%) in our study cohort was somewhat higher than the overall seroprevalence found in healthcare workers of other german university hospitals in Essen (1.6%, n = 316) and Hannover (1.0–2.0%, n = 217), and in a neurological center in southern Germany (2.9%, n = 406), respectively [4–6].

These findings might be due to more high risk exposure of physicians in intensive care units and emergency medical services compared to general hospital staff, especially since 73% of our study participants had treated COVID-19 patients and the majority were younger physicians, i.e. registrars with a mean working experience of 5.3 years. These usually are the frontline workforce and have frequent patient interaction. In line with our data, a higher seroprevalence of 5.4% occured in a high-risk group of HCW at a university hospital who had daily contact to COVID-19 patients [4].

In studies on healthcare personnel working in emergency departments in the United States the seroprevalence was up to 7.6% in Tennesse [7] and between 5.9% - 11.5% depending on the test procedure in an emergency department in Utah [8]. In a tertiary reference hospital for

Infectious Diseases in Brussels during the first peak of the pandemic the SARS-CoV-2 carriage and seroprevalence showed an overall infection rate of 12.6% [8] which is almost similar to the infection rate of 11.2% in a Spanish reference hospital for Covid-19 patients [9]. In a review of 11 studies including 119,216 Covid-19 patients the proportion of healthcare workers tested positive was 10.1%, with the lowest rate of 4.2.% in China and the highest rate of 17.8% in the United States [10]. An interesting study with a large number of participants was published from the UK. Approx. 10.000 healthcare staff were tested voluntarily with RT-PCR and immunoassays for antibodies. The overall positivity was higher compared to our data, i.e. 11.2%. The authors provide a detailed analysis and in line with our results staff with a confirmed household contact had the highest risk for Covid-19 with an OR of 4.82. Interestingly, physicians faced a lower risk compared to other healthcare staff which is also in line with the relatively low rate of infection in our participants [11].

Thus, the overall infection rate of intensive care and emergency physicians in our study was lower compared to similar at-risk groups in other countries. This finding may be representative for a health-care setting were hospital capacities have not been overwhelmed and where personal protective equipment was widely available and used.

In our study approx. 39% of those tested positive were unaware of their infection. This is in line with other studies [6] and underlines that a large number of SARS-CoV-2 infections remain oligo- or asymptomatic. Of note, the incidence of severe Covid-19 disease in healthcare workers was significantly lower compared to the overall group of Covid-19 patients (9.9% vs. 29.4%; p<0.001) [10]. On the one hand these mild disease courses may help to overcome fears of healthcare personnel during daily work. On the other hand the asymptomatic courses may increase the risk of spreading the disease. Thus, this finding may be an argument for routine testing of staff in high-risk areas.

We found that approx. one third of our participants had traveled abroad in 2020. These travelers visited 57 countries but the majority stayed in Europe. Many participants visited high-risk and so-called COVID-19 hotspot areas but we detected no statistically significant correlation between travel activities and risk for SARS-CoV-2 infection.

We found that 17% of our participants had a febrile respiratory infection in 2020. However, only 4.5% of all febrile respiratory infections were related to a proven SARS-CoV-2 infection. This finding highlights that other causes for respiratory infections are common and need to be considered in the diagnostic work-up. However, the upcoming autumn and winter season will cause an increase of respiratory infections. It is well known that many physicians–as well as other healthcare personnel–continue to work with mild or even moderate respiratory infections. During the present pandemic situation this attitude is unacceptable because the risk to infect fellow HCW and vulnerable patients is intolerable. Thus, it will be mandatory to use highly sensitive and specific diagnostic test procedures to rapidly differentiate a SARS-CoV-2 infection from other respiratory infections. A risk of understaffing during these critical times must be taken into account and interventions to compensate are mandatory.

We found a statistically significant correlation between the risk of SARS-CoV-2 positivity of physicians and contact to family members with COVID-19. It is remarkable that despite well defined risk factors in hospitals and during work with COVID-19 patients the highest risk for infection of German healthcare professionals still seems to be related to domestic life. If we compare our data to results from other countries, however, one must consider that PPE was widely available and that the healthcare system worked within normal limits.

In general, a sufficient supply of PPE was available in the hospitals. We specifically asked for the availiability of face masks with high filtering capacity which can protect from airborne infection [12, 13]. We found that such masks were not available to 13% of participants at some point in time during the pandemic. The same observation occurred for eye protective devives

which were unavailable to 18% at some point in time. However, we found no correlation between the risk of SARS-CoV-2 transmission and lack of PPE. The use of such devices is most important during high risk procedures such as intubation, bronchoscopy, non-invasive ventilation or nasal high-flow oxygen. In situations were PPE was not available, physicians might have used other techniques for protection like wearing double or triple layers of conventional masks and advanced technical support for intubation [14]. The application of such solutions was not asked for in our questionnaire. Nevertheless, it is worrisome that a lack of adequate PPE for frontline healthcare workers occurred. In countries where the COVID-19 disease occurred early, high rates of SARS-CoV-2 related infection and associated deaths of healthcare workers were observed. These were partly explained by inadequate access to PPE [15, 16].

Finally, we found that 51% of our participants had been vaccinated against seasonal influenza during the 2019/2020 season. The vaccination rate of healthcare workers is notoriously low, despite public recommendations. For the upcoming autumn/winter season, a higher vaccination rate in healthcare workers is desirable to reduce the number of influenza-associated respiratory diseases. Data from 223 healthcare trusts in the United Kingdom showed that a 10% increase in vaccination rate was associated also with a 10% fall in sickness absence rate [17]. This way, a shortage of staff might be mitigated during the critical winter period.

## Strengths of this study

An important strength of this study is the concentration on the homogenous group of intensive care and emergency physicians working in high-risk areas. Most studies on healthcare personnel include staff from different hospital areas thus often not focussing on those with the highest infection risk.

Another strength is that participants came from all parts of Germany and from hospitals of different levels. This way, we present a broad view on the German hospital landscape. Most important, almost all participants had been in contact with COVID-19 patients.

## Limitations of this study

We recruited only physicians attending these voluntary courses and thus may have focused on a highly motivated group but this should have no impact on the seroprevalence. Not every participant answered every question, however, the lowest quote was 497/516. Thus, our conclusions rely on a high number of answers. Our study population was slightly younger compared to the average of German hospital physicians. However, these younger physicians always worked in frontline situations and thus have a high infection risk. We did not ask for the number/duration of high exposures like intubation or other aerosol generating procedures because it is unlikely to obtain reliable data in a retrospective survey. We observed a ratio of male to female of 1:1 while in Germany approx. 63% of all physicans are female. Therefore, our study population is not completely representative with regard to gender proportion but we do not believe that it has an impact on the conclusion.

Finally, it is unclear whether and to what extent antibodies against SARS-CoV-2 persist or whether there is a rapid decay within 3 months after the infection [18, 19]. We reduced the risk of underdiagnosis by also asking for results of RT-PCR tests done in our study group. In fact, we identified two physicans with negative antibody status after proven SARS-CoV-2 infection. Additional cases of SARS-CoV-2 infection might have been missed due to seronegativity, oligo- or asymptomatic disease presentation without RT-PCR testing or because the study period was rather long and some participants may not remember their illness if it occurred several months ago.

## Conclusion

The overall positive rate for SARS-CoV-2 infection in German intensive care and emergency physicians evaluated by seroprevalence of antibodies and results of RT-PCR testing was 3.5%, and 39% of those infected were unaware. The only significant risk factor for SARS-CoV-2 infection in this study population was contact to COVID-19 positive relatives in the domestic environment. The overall infection rate of German physicians working in high-risk areas was low compared to other countries and settings. This finding may be explained by the fact that the German health care system was not overwhelmed by the first wave of the SARS-CoV-2 pandemic and that personal protective equipment was widely available.

## Author Contributions

**Conceptualization:** Detlef Kindgen-Milles, Timo Brandenburger, Corvin Cleff, Kian Moussa-zadeh, Jörg Timm, Dietmar Wetzchewald.

**Data curation:** Julian F. W. Braun, Corvin Cleff, Kian Moussazadeh, Jörg Timm, Dietmar Wetzchewald.

**Formal analysis:** Detlef Kindgen-Milles, Corvin Cleff, Ingo Mrosewski, Dietmar Wetzchewald.

**Investigation:** Kian Moussazadeh.

**Methodology:** Kian Moussazadeh, Ingo Mrosewski, Jörg Timm, Dietmar Wetzchewald.

**Project administration:** Detlef Kindgen-Milles, Dietmar Wetzchewald.

**Resources:** Julian F. W. Braun, Dietmar Wetzchewald.

**Supervision:** Dietmar Wetzchewald.

**Validation:** Detlef Kindgen-Milles, Julian F. W. Braun, Dietmar Wetzchewald.

**Writing – original draft:** Detlef Kindgen-Milles, Dietmar Wetzchewald.

**Writing – review & editing:** Timo Brandenburger, Corvin Cleff, Kian Moussazadeh, Ingo Mrosewski, Jörg Timm, Dietmar Wetzchewald.

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
