## [Decision Letter · Decision Letter 0]

25 Jan 2021

PONE-D-20-39396

Prevalence of SARS-COV-2 positivity in 516 german intensive care and emergency physicians studied by seroprevalence of antibodies National Covid Survey Germany (NAT-COV-SURV)

PLOS ONE

Dear Dr. Kindgen-Milles,

Thank you for submitting your manuscript to PLOS ONE. After careful consideration, we feel that it has merit but does not fully meet PLOS ONE’s publication criteria as it currently stands. Therefore, we invite you to submit a revised version of the manuscript that addresses the points raised during the review process.

We look forward to receiving your revised manuscript.

Kind regards,

Ruslan Kalendar, PhD

Academic Editor

PLOS ONE

Journal Requirements:

2. Please provide your main results (both significant and non-significant) in a table, and include all numerial p-values.

3.We note that you have indicated that data from this study are available upon request. PLOS only allows data to be available upon request if there are legal or ethical restrictions on sharing data publicly. For information on unacceptable data access restrictions, please see http://journals.plos.org/plosone/s/data-availability#loc-unacceptable-data-access-restrictions.

4.Thank you for stating the following in the Declarations Section of your manuscript:

"This study was funded by internal funding of Working Group Intensive Care Medicine,

Arnsberg, Germany"

 "The authors received no specific funding for this work"

5.Thank you for stating the following in the Competing Interests section:

"The authors have declared that no competing interest exist"

We note that one or more of the authors are employed by a commercial company: MDI Limbach Berlin GmbH, Berlin

6.We note that Figure 1 in your submission contain map images which may be copyrighted. All PLOS content is published under the Creative Commons Attribution License (CC BY 4.0), which means that the manuscript, images, and Supporting Information files will be freely available online, and any third party is permitted to access, download, copy, distribute, and use these materials in any way, even commercially, with proper attribution. For these reasons, we cannot publish previously copyrighted maps or satellite images created using proprietary data, such as Google software (Google Maps, Street View, and Earth). For more information, see our copyright guidelines: http://journals.plos.org/plosone/s/licenses-and-copyright.

a) You may seek permission from the original copyright holder of Figure 1 to publish the content specifically under the CC BY 4.0 license. 

7. Please include your tables as part of your main manuscript and remove the individual files. Please note that supplementary tables (should remain/ be uploaded) as separate "supporting information" files.

8. Your ethics statement should only appear in the Methods section of your manuscript. If your ethics statement is written in any section besides the Methods, please delete it from any other section.

Reviewers' comments:

Reviewer's Responses to Questions

**Comments to the Author**

1. Is the manuscript technically sound, and do the data support the conclusions?

Reviewer #1: Yes

Reviewer #2: Yes

Reviewer #3: Yes

2. Has the statistical analysis been performed appropriately and rigorously? 

Reviewer #1: Yes

Reviewer #2: Yes

Reviewer #3: Yes

3. Have the authors made all data underlying the findings in their manuscript fully available?

Reviewer #1: Yes

Reviewer #2: No

Reviewer #3: No

4. Is the manuscript presented in an intelligible fashion and written in standard English?

Reviewer #1: Yes

Reviewer #2: Yes

Reviewer #3: Yes

5. Review Comments to the Author

Reviewer #1: 

The article is overall well written and may be useful for scientific community and physicians.

I only have few comments.

In the method section, it is not clearly mentioned which participants have been tested by RT PCR. While in the results section, it is stated that only 222 participants have been tested by RT PCR and/or antibodies. I think it should be more explicitly explained.

Then there is a major published article on this issue that should be discussed in the discussion section. Since it included 10 034 healthcare workers in the UK and the comparison is interesting.

DOI: https://doi.org/10.7554/eLife.60675

Reviewer #2: 

This is a point prevalence study of SARS CoV2 seropositive rate among 516 Physicians, who were recruited over a 2 week period between June 16th and Jul 2nd 2020.

The overall Seropositivity rate was 3.5% (18/516). The authors report a significant correlation between SARS-Cov2 seropositivity and a history of contact with SARS-CoV-2 family members.

I have the following comments and questions for the authors.

1. Results should be shown in table format for ease of reading and interpretation. Would be helpful to show (in table form) the seropositive rates in specific subgroups e.g. those who cared for COVID patients, those with a positive family member, those with hx of febrile respiratory tract infection.

2. The authors state that “…11/18 (61%) were aware of their infection”. However, only 2 participants had previous positive PCR tests…how did the remaining 9 participants know about their infection? Through symptoms alone?

3. Was there any association between treating COVID positive patients and Seropositivity?

4. “A febrile resp tract infection after Jan 1 occurred in 87/499 (17%)…….but only 4/87 of those with a febrile resp infection were SARS-CoV2 positive, while 14/18 of positive patients had no febrile resp infection”. Suggest re-word to:

87/499 (17%) participants reported a febrile resp tract infection after Jan 1st. Among these, <5% were seropositive for SARS Cov2.

Among the 16 participants who were SARS Cov2 positive, only 2/18 reported a previous febrile resp infection after Jan 1st.

5. Hx of vaccination against seasonal influenza is not really relevant here, and can be removed. (why would previous flu vaccination affect serology results?)

6. A limitation of the study is the long period (Jan to Jun) that participants were required to recall their symptoms, contact histories etc. For eg, the low number (2/18) of positive participants who reported previous febrile illness may be due to the fact that many participants could not remember their illness if it occurred several months ago.

Reviewer #3: 

This study has been well conducted and the statistical analysis is appropriate. More sophisticated techniques, including multivariable logistic regression, or accounting for geographic differences might have been considered, but due to the low prevalence and sparsity of cases is not necessary/possible. The low prevalence, given high exposure to COVID-19 patients, is slightly surprising, as is the finding that domestic risk is highest The authors draw balanced conclusions on this. I have no recommendations for editing.

6. PLOS authors have the option to publish the peer review history of their article (what does this mean?). If published, this will include your full peer review and any attached files.

Reviewer #3: **Yes: **Ross Harris

---

## [Author Response · Author response to Decision Letter 0]

5 Mar 2021

PONE-D-20-39396 Response to Reviewers

Dear Editor, Dear Reviewers,

we appreciate very much your comments which help us to improve this manuscript. We adressed all requirements as follows:

To the Editor

1) We checked the style requirements and renamed all files, if necessary

2) We provided a table (Table 3) which shows relevant results regardless of significance. We included p-values.

3) According to german federal and state jurisdiction (DGSVO, Datenschutzgrund-verordnung) data are protected and cannot be shared publicly. The approval by the committee of medical ethics at Heinrich-Heine University Duesseldorf (*2020-1034) did not allow transfer of data outside of the university. However, data can be made available upon request. For any data request researchers may contact: (1) Committee of Medical Ethics of Heinrich-Heine University, Medical Faculty Heinrich-Heine University Duesseldorf, Moorenstr. 5, D-40225 Duesseldorf; ethikkommission@med.uni-duesseldorf.de, or (2) the data protection commissioner of the university hospital: Datenschutzbeauftragter, University Hospital Duesseldorf, Moorenstr. 5, D-40225 Duesseldorf; datenschutz@med.uni-duesseldorf.de

4) We removed the funding information from the Acknowledgement section. There was no specific funding. All expenses including laboratory analysis were paid by regular institutional budget.

5) Dr. Ingo Mrosewski is employed by the MDI Limbach Berlin GmbH, a commercial clinical laboratory. The MDI Limbach GmbH did not play a role in the study design, data collection and analysis, decision to publish, or preparation of the manuscript. It did not provide specific financial support but all costs for laboratory analysis were paid for by the Dept. of Emergency Medicine at University of Herdecke. This affiliation does not alter our adherence to PLOS ONE policies on sharing of data and materials. All other authors have nothing to declare

6) We removed figure 1 from this submission. We considered presenting the data in a table. However, when thoroughly analysing the information, the figure as well as table would just show that participants came from almost all parts of Germany, with the highest number coming (as expected) from the most densily populated areas, and no relation between rate of positivity and region. We mentioned that in the results section stating that participants came from all over germany and naming the contribution oft he largest federal states. For the sake of brevity, we removed the figure withour adding a new table. 

7) We included table 3 in the manuscript as well. There are no supplementary files.

8) The ethics statement has been removed from the acknowledgement section.

Reviewer # 1:

1) In the method section, it is not clearly mentioned which participants have been tested by RT PCR. While in the results section, it is stated that only 222 participants have been tested by RT PCR and/or antibodies. I think it should be more explicitly explained.

We clarified this aspect in the methods section (page 4):

All participants answered a clinical questionnaire and from all participants 8 ml blood were drawn for SARS-CoV-2 antibody detection. All questionnaires and blood samples were given matching numbers and blinded. In some participants RT-PCR testing had been performed independent from this study. Participants were asked whether such tests had been performed and to provide the results of this tests, i.e. positive or a negative result. 

We clarified this aspect also in the results section (page 6):

We found antibodies against the N protein of SARS-CoV-2 in 16/516 (3.1%) participants. SARS-CoV-2 tests via nasopharyngeal swab and RT-PCR and/or antibody detection independent from this study, i.e. at a time prior to participation in this study, had been performed in 222/516 (43%).

2) Then there is a major published article on this issue that should be discussed in the discussion section. Since it included 10 034 healthcare workers in the UK and the comparison is interesting.

We added the article published by Eyre et al. and discussed it (Discussion, page 9)

An interesting study with a large number of participants was published from the UK. Approx. 10.000 healthcare staff were tested voluntarily with RT-PCR and immunoassays for antibodies. The overall positivity was higher compared to our data, i.e. 11.2%. The authors provide a detailed analysis and in line with our results staff with a confirmed household contact had the highest risk for Covid-19 with an OR of 4.82. Interestingly, physicians faced a lower risk compared to other healthcare staff which is also in line with the relatively low rate of infection in our participants 11.

Reviewer #2:

1. Results should be shown in table format for ease of reading and interpretation. Would be helpful to show (in table form) the seropositive rates in specific subgroups e.g. those who cared for COVID patients, those with a positive family member, those with hx of febrile respiratory tract infection.

We added a table (table 3) showing these results, i.e. data for those with a household contact, a stay abroad, a feverish resp. infection. We added also data on positivity rate versus availability of personal protective equipment. However, 86 % of participants worked in high risk-areas and 73% had a confirmed contact with Covid-19 patients. Therefore, we believe that almost everybody had a contact to SARS-CoV-2 positive patients, either knowingly or not. In the light of the low positivity rate, we think it is not meaningful to show separate data for these subgroups.

2. The authors state that “…11/18 (61%) were aware of their infection”. However, only 2 participants had previous positive PCR tests…how did the remaining 9 participants know about their infection? Through symptoms alone?

Thank you for this point. Here, our manuscript was unclear. The two positive RT-PCR tests were mentioned by participants in whom antibody testing was negative. Thus, the previous RT-PCR tests detected these two particpants. In other participants, PCR tests were done as well but provided no additional information because these particpants were detected by our antibody testing. We clarified this aspect in the Abstract section and in the results section

Page 2

We recruited 516 physicans from all parts of Germany, 445/516 (86%) worked in high risk areas, and 379/516 (73%) had treated patients with COVID-19. The overall positive rate was 18/516 (3.5%), 16/18 (89%) had antibodies against SARS-COV-2, another 2 reported previous positive RT-PCR results although antibody testing was negative.

Page 6

Of those with positive antibodies or a positive RT-PCR 11/18 (61%) were aware of their infection – mostly due to previous RT-PCR testing unrelated to this study - and 7/18 (39%) were not aware. 

3. Was there any association between treating COVID positive patients and Seropositivity?

see #1

4. “A febrile resp tract infection after Jan 1 occurred in 87/499 (17%)…….but only 4/87 of those with a febrile resp infection were SARS-CoV2 positive, while 14/18 of positive patients had no febrile resp infection”. Suggest re-word to:

87/499 (17%) participants reported a febrile resp tract infection after Jan 1st. Among these, <5% were seropositive for SARS Cov2.

Among the 16 participants who were SARS Cov2 positive, only 2/18 reported a previous febrile resp infection after Jan 1st.

DONE, page 7

5. Hx of vaccination against seasonal influenza is not really relevant here, and can be removed. (why would previous flu vaccination affect serology results?)

We agree that previous flu vaccination should not affect infection rate with SARS-CoV-2 or serology results in healthcare staff. However, it was easy to obtain this information and we think that the additional information is worth to be given to the readers for a number of reasons. We have discussed this part on page 11. 

At present, there is an ongoing discussion on the relevance of vaccination. Surprisingly, in some countries including Germany a number of healthcare staff is hesitant to be vaccinated against SARS-CoV2 despite clear data which speak in favour of the vaccination. The same observation has been made in recent years regarding vaccination against seasonal flu and data here show large positive effects which can be achieved with higher vaccination rates. Therefore, in the light of this ongoing debate, we believe every positive aspect of vaccination is worth tob e mentioned. We would be delighted if this (small) part of the manuscript could be accepted.

6. A limitation of the study is the long period (Jan to Jun) that participants were required to recall their symptoms, contact histories etc. For eg, the low number (2/18) of positive participants who reported previous febrile illness may be due to the fact that many participants could not remember their illness if it occurred several months ago

We have mentioned this limitation in the Limitation section on page 12:

Additional cases of SARS-CoV-2 infection might have been missed due to seronegativity, oligo- or asymptomatic disease presentation without RT-PCR testing or because the study period was rather long and some participants may not remember their illness if it occurred several months ago. .

Reviewer #3: 

Thank you very much for your assessment

---

## [Editor Report · Decision Letter 1]

8 Mar 2021

Prevalence of SARS-COV-2 positivity in 516 german intensive care and emergency physicians studied by seroprevalence of antibodies National Covid Survey Germany (NAT-COV-SURV)

PONE-D-20-39396R1

Dear Dr. Kindgen-Milles,

We’re pleased to inform you that your manuscript has been judged scientifically suitable for publication and will be formally accepted for publication once it meets all outstanding technical requirements.

Kind regards,

Ruslan Kalendar, PhD

Academic Editor

PLOS ONE

---

## [Editor Report · Acceptance letter]

31 Mar 2021

PONE-D-20-39396R1 

Prevalence of SARS-COV-2 positivity in 516 german intensive care and emergency physicians studied by seroprevalence of antibodiesNational Covid Survey Germany (NAT-COV-SURV) 

Dear Dr. Kindgen-Milles:

I'm pleased to inform you that your manuscript has been deemed suitable for publication in PLOS ONE. Congratulations! Your manuscript is now with our production department. 

Kind regards, 

on behalf of

Prof. Ruslan Kalendar 

Academic Editor

PLOS ONE